# Morphological Analysis of Poly(4,4′-oxydiphenylene-pyromellitimide)-Based Organic Solvent Nanofiltration Membranes Formed by the Solution Method

**DOI:** 10.3390/membranes12121235

**Published:** 2022-12-07

**Authors:** Tatyana E. Sukhanova, Andrey L. Didenko, Ilya L. Borisov, Tatyana S. Anokhina, Aleksey G. Ivanov, Anna S. Nesterova, Ilya A. Kobykhno, Alexey A. Yushkin, Vladimir V. Kudryavtsev, Alexey V. Volkov

**Affiliations:** 1S.V. Lebedev Institute of Synthetic Rubber, Gapsal’skaya ul. 1, 198035 St. Petersburg, Russia; 2A.V. Topchiev Institute of Petrochemical Synthesis, Russian Academy of Sciences, 29 Leninsky Prospekt, 119991 Moscow, Russia; 3Institute of Macromolecular Compounds, Russian Academy of Sciences, Bolshoy pr. V.O., 31, 199004 St. Petersburg, Russia; 4Laboratory of Polymer Composites, Peter the Great St. Petersburg Polytechnic University, Polytechnicheskaya, 29, 195251 St. Petersburg, Russia

**Keywords:** poly(4,4′-oxydiphenylene–pyromellitimide), organic solvent nanofiltration, membrane, copoly(urethane-imide)s, morphology, multiblock (segmented) copolymers, solution method

## Abstract

Poly-(4,4′-oxydiphenylene) pyromellitimide or Kapton is the most widely available polyimide with high chemical and thermal stability. It has great prospects for use as a membrane material for filtering organic media due to its complete insolubility. However, the formation of membranes based on it, at the moment, is an unsolved problem. The study corresponds to the rediscovery of poly(4,4′-oxydiphenylene–pyromellitimide)-based soluble copoly(urethane-imides) as membrane polymers of a new generation. It is shown that the physical structure of PUI films prepared by the solution method becomes porous after the removal of urethane blocks from the polymer, and the pore size varies depending on the conditions of thermolysis and subsequent hydrolysis of the membrane polymer. The film annealed at 170 °C with a low destruction degree of polycaprolactam blocks exhibits the properties of a nanofiltration membrane. It is stable in the aprotic solvent DMF and has a Remasol Brilliant Blue R retention coefficient of 95%. After the hydrolysis of thermally treated films in acidic media, ultrafiltration size 66–82 nm pores appear, which leads to an increase in the permeate flow by more than two orders of magnitude. This circumstance provides opportunities for controlling the membrane polymer structure for further optimization of the performance characteristics of filtration membranes based on it. Thus, we proposed a new preparation method of ultra- and nanofiltration membranes based on poly(4,4′-oxydiphenylene–pyromellitimide) that are stable in aprotic solvents.

## 1. Introduction

The rapid growth of the world’s population, the increasing mobility of movement, and globalization have demonstrated the vulnerability of modern society to the rapid, cross-border spread of new diseases (COVID-19, SARS, H5N1, H1N1 influenza, etc.). Synthesis of antivirals involves the use of bio- and/or metalorganic catalysts, a number of organic solvents, includes 4 to 20 different reactions requiring separation of a catalyst from the reaction mixture (including for reuse), removal and replacement of solvents, concentration and isolation of target and by-products, meanwhile many stages of the synthesis of several precursors are performed at temperatures of 80–250 °C [1,2,3].

These separation problems in organic synthesis can be effectively solved by applying baromembrane processes—organic solvent nanofiltration (OSN) [4,5].

Polyimides are considered to be the most promising group of polymeric heat-resistant materials for this purpose [6]. Existing industrial asymmetric polyimide nanofiltration membranes of the DuraMem series (Evonik) are produced based on the solvent technology by phase inversion (from aprotonic solvents) and subsequent cross-linking of the membrane to increase its stability in organic solvents. For this purpose, commercial soluble polyimides Lenzing P84 [7,8] Matrimid [9,10,11], and PMDA/ODA [12] are used.

The problem of membrane stability can be solved by using insoluble aromatic polyimides [13], which, in turn, requires a process of membrane molding from a soluble prepolymer (polyamide acid—PA) [14] followed by thermal or chemical imidation of PA to the corresponding polyimide. 

However, this approach has a number of disadvantages: instability of PA in time and in contact with water vapor, difficulty in controlling the porous structure of the membrane, and difficulties in forming hollow fiber membranes from PA solutions (high viscosity of the solution, long precipitation process). A special place in the range of modern thermoplastics is occupied by multiblock (segmented) copolymers containing rigid imide blocks and flexible blocks of aliphatic esters or ethers–copolyetherimides (PEI) or copolyuretanes (PUI) [15,16]. The works carried out in the field of synthesis, and the property research of thermoplastics are of interest to modern science and technology, which is in urgent need of thermal-resistant and highly heat-resistant elastic polymer materials. Such materials are quite promising for application in airspace technique, shipbuilding and car industry, microelectronics, and membrane technology [17,18,19,20,21,22,23,24,25,26,27]. 

Poly-(4,4′-oxydiphenylene) pyromellitimide is the most affordable polyimide produced by industry in the form of an electrically insulating film. It is well known that polyimides as membrane materials are characterized by high selectivity of separable substances at low productivity. It seems interesting to develop methods for obtaining a polyimide film suitable for nano- and ultrafiltration of process fluids. That is, to make the necessary changes to the poly-(4,4′-oxydiphenylene) pyromellitimide film technology for this.

Due to the differences in the reactivity of the imide and urethane links with respect to thermolysis and hydrolysis in acidic media, we previously showed selective degradation of polycaprolactam blocks in the synthesized copoly(urethane-imide) [19]. However, from the aspect of membrane properties, it is important that transport channels forming the porous membrane structure are formed in place of the destructive flexible blocks. For this purpose, in the present study, we carried out selective destruction of the synthesized copoly(urethane-imide) films by thermolysis or thermolysis combined with destruction (hydrolysis) in acidic media and investigated the morphological and transport properties of the obtained (formed) materials. The morphology of the obtained films was studied in detail by the atomic force microscope (AFM). AFM, in combination with control software and AFM image processing tools, is designed to measure and analyze micro- and submicro relief of objects surfaces on nanometer range with high resolution. AFM is an indispensable tool in the study of the structure and morphology, as well as local mechanical and tribological properties on the submicro- and nano-level.

## 2. Materials and Methods

### 2.1. The Object under the Study

A complex architecture copolymer of [(imide)n-(urethane)]m type, including units of poly-(4,4′-oxydiphenylene) pyromellitimide and units of bisurethane obtained on the basis of polycaprolactone and 2,4-toluene diisocyanate, was chosen as an object of our research. This choice makes it possible to predict the stability in corrosive environments of poly-(4,4′-oxydiphenylene) pyromellitimide modified by the inclusion of urethane links in its structure. 

A complex architecture copolymer of the [(imide)_n_-(urethane)]_m_ type, including blocks of poly-(4,4′-oxydiphenylene) pyromellitimide and blocks of bisurethane derived from polycaprolactone and 2,4-toluene diisocyanate, was chosen as an object of the research. This choice makes it possible to predict the stability in corrosive environments of poly-(4,4′-oxydiphenylene) pyromellitimide modified by the inclusion of urethane links in its structure.

### 2.2. Materials

The copoly(urethane-imide) chosen as the object of study was obtained using the synthesis method described in our initial studies [28,29,30,31,32,33,34]. In this case, as monomers, we used pyromellitic anhydride with a melting point (Tm) ~283–286 °C, Sigma-Aldrich Co. LLC, St. Louis, MO, USA; polycaprolactone with molecular weight Mn = 2000 with melting point (Tm) ~50 °C, Sigma-Aldrich Co. LLC; 2,4-toluene diisocyanate with melting point (Tm) ~20–22 °C, Sigma-Aldrich Co. LLC; 4,4′-diaminodiphenyl ether with melting point (Tm) ~188–192 °C, Sigma-Aldrich Co. LLC. N,N-dimethylacetamide, Vecton Co. LLC, London, UK, Russia was used as a solvent.

### 2.3. Synthesis of coPUIs

The method used to synthesize the target copolymer involves the copolycondensation of 4,4-diaminodiphenyl ether in solution of N,N-dimethylacetamide with pyromellitic anhydride and a macromonomer with end anhydride groups, which is obtained from polycaprolactone, 2,4-toluene diisocyanate and pyromellitic anhydride. This method is thoroughly described in our previous work [19]. In this work, its structure was proved. 

The polymer formula is presented below (Figure 1):

where *n* = 1, m = 10.

NMR ^1^H prepolymer (DMSO-d_6_) *δ*, figures: 9.91, 9.62, 9.12, 8.87, 8.38, 7.98–6.55, 3.99, 3.66, 2.29, 2.15, 2.11, 2.08, 1.54, 1.31.

The intrinsic viscosity of the prepolymer was [η]= 1, 1 × 100 cm^3^ g^−^^1^.

Hydrolysis and thermolysis of copoly(urethane-imide) were described in our previous work [19].

### 2.4. PUI Films Obtaining

Film samples of copolymers were obtained by molding on hydrophobic glass plates. Films were heated at the following temperature modes: 12 h at 80 °C, 1 h at 100 °C, 1 h at 120 °C, 1 h at 140 °C, 2 h at 170 °C. This resulted in complete thermal conversion of the copoly(urethane amic acid) into the target copoly(urethane-imide). Copoly(urethane-imide) films fixed on glass plates were placed in a furnace and heated stepwise. The annealing temperature varied from 170 °C to 370 °C with a duration of 0.5 h to 2 h. After annealing, the films were removed from the plates.

In order to intensify the destruction, the samples of copoly(urethane-imide) films heated at different temperature modes were hydrolyzed in solutions of acetic, hydrochloric, and trifluoroacetic acids. The concentrations of acids in the hydrolyzing solutions were varied in the range of 10 to 90 vol.%. Mixtures of acids were also used for etching. Distilled water was used as a diluent; its concentration varied from 10 to 90 percent in the hydrolysis solution. Another parameter monitored in this process was the time of film etching. It is worth noting that in most of the hydrolyzed blends, the films were destroyed and became unsuitable for further studies. Finally, three hydrolysis compositions were chosen, after processing with which the films remained self-supporting: the blend of concentrated acetic acid and hydrochloric acid taken in ratio of 90:10 (vol. %) (time of hydrolysis blend action is 24 h); the blend of acetic acid and concentrated hydrochloric acid in the ratio of 10:90 (vol. %) (time of hydrolysis blend action is 12 h); the blend of trifluoroacetic acid and water in ratio of 50:50 (vol. %) (time of hydrolysis blend action is 24 h). The samples were stored in the blend at room temperature; after that, they were washed with distilled water in the washing bath until a neutral reaction was achieved. Before further studies, the films were dried in the open air. Designations of the films obtained in the work are given in Table 1.

### 2.5. AFM Investigation

The surface morphology of the synthesized copolymer films was investigated by the AFM method on the AFM device Nanosurf FlexAFM (produced by Nanosurf AG, Switzerland). The shooting was carried out in dynamic mode using standard *Tap150Al-G* cantilever with a radius of tip curvature of less than 10 nm and stiffness of cantilever 5 N/m (produced by Budget Sensor, Bulgaria). Overscan is 20%. For each sample, a survey was carried out with a field of 20 μm *×* 20 μm and 1 μm *×* 1 μm. For each field of the survey, the results of *Z*-axis are presented—the topography of the surface in nm and the Amplitude—which allows you to better assess the topography of the surface at large magnifications.

The roughness, coefficient of friction, and friction force in nano- and microscale of samples were evaluated. The surface roughness was estimated in scanning areas of 20 μm × 20 μm and 1 μm × 1 μm.

The measurements were performed using a FlexAFM atomic force microscope (manufacturer Nanosurf AG, Liestal, Switzerland). The measurements were taken in dynamical mode using a Tap150Al-G cantilever (manufacturer Budget Sensor, Sofia, Bulgaria, resonant frequency 150 kHz, force constant 5 N/m).

A field of 20 μm × 20 μm and 1 μm × 1 μm microns was taken for each sample. The overscan is 20%.

### 2.6. X-ray Diffraction (XRD)

X-ray diffraction (XRD) studies of films were performed on a Rigaku diffractometer, Japan. Experimental diffractograms were obtained using an X-ray source with a rotating copper anode Rotaflex RU-200. The operating mode of the source was 50 kV–100 mA. The source was equipped with a horizontal wide-angle Rigaku D/Max-RC goniometer and a secondary graphite monochromator (the wavelength λ of the monochromatic radiation was 1.542 Å). Film samples were fixed on aluminum frames; in this case, scanning was performed in the reflection mode. It should be noted that with the X-ray wavelength used in the reflection mode, the beam scanned the entire depth of the films.

### 2.7. Filtration Experiments

The membranes were tested in nanofiltration of DMF in dead-end cells at 20 bar. The mixture in the cells was constantly stirred with magnetic stirrers. The permeate flow was determined by the gravimetric method.

A liquid receiver was installed at the outlet of the cell. The mass of permeate passing through the membrane during the experiment was measured on a Sartorius laboratory balance with a measurement error of 0.001 g. Membrane performance was characterized by liquid permeation, *P*(1):(1)$$P=\frac{m}{S\;\cdot \;\Delta t\;\cdot \;\Delta p}$$
where *m* is the weight of permeate (kg) passed through a membrane with an area *S* (m^2^) over a time Δ*t* (h), Δ*p* is the pressure drop.

The optical density of solutions was measured with PE-5400UF spectrophotometer (PromEcoLab, Saint Petersburg, Russia). First, the concentrations of model compounds (dyes) in the feed and permeate were determined using the calibration curve; after the rejection *R* (%) was calculated and used to evaluate the separation characteristics of the membrane:(2)$$R=\left(1-\frac{C_{p}}{C_{0}}\right)\cdot 100\%$$

*C_p_* and *C*_0_ are the dye concentrations in the permeate and feed, respectively.

### 2.8. Sorption and Swelling Studies

Sorption measurements were conducted by placing membrane samples in selected solvents for several days after preliminary weighing of the samples. After soaking in solvents, the membranes were taken out, the excess solvent from the surface of the samples was removed with filter paper, and the membranes were weighed daily. The measurements were stopped when the membrane mass did not change for 2 days. The sorption (*K*_s_) was calculated by the following Equation (3):(3)$$K_{s}=\frac{\left(m_{1}-m_{0}\right)}{m_{0}}$$
where *m*_0_ and *m*_1_ are the masses of the dry and swollen samples, respectively.

In order to measure the swelling degree, rectangular membrane samples were placed in solvents for a day after measuring the geometric dimensions (length, width, and thickness) of the sample. After soaking the samples in solvents, their geometric dimensions were again measured. The degree of swelling of cellulose in *S_D_* liquids was determined by Formula (4):(4)$$S_{D}=\frac{d_{1}\cdot d_{2}\cdot l-d_{10}\cdot d_{20}.l_{0}}{d_{10}\cdot d_{20}.l_{0}}\cdot 100\%$$
where *d*_10_, *d*_20_, *d*_1_, *d*_2_, *l*_0_, and *l* are the length, width, and thickness, respectively, of the initial and swollen samples. Index 0 denotes dry sample dimensions.

The pore size distribution (PSD) was measured by a liquid-liquid displacement porometry (LLDP) [14] using porometer POROLIQ 1000 ML (Porometer, APTCO TECHNOLOGIES NV, Nazareth, Belgium).

## 3. Results and Discussion

### 3.1. AFM Investigation

The surface of the synthesized copoly(urethane-imide) films was examined by atomic force microscopy (AFM) using 1 μm × 1 μm and 20 μm × 20 μm scan matrices.

Appendix A shows AFM images of fragments of the upper (free) surface of the PUI polymer film (20 μm × 20 μm scan matrix) obtained at different stages of the sample heating from 170 to 370 °C. It can be seen that all samples have the so-called technological surface geometry. Numerous scratches and grooves are observed on it, and the surface is textured, apparently, in the direction of movement of the doctor blade used to obtain the film, as evidenced by regular parallel straight stripes formed by irregularly shaped particles. These stripes are most clearly observed in samples heated to 170 and 200 °C. 

In higher resolutions, AFM images of the fine structure of the film surface heated at 170 °C (Figure 1a and Appendix A, scan matrix 1 μm × 1 μm), a fine-grained morphology with grain sizes from 20 to 50 μm is clearly visible. Small gaps of about 10–20 μm in diameter can be seen between the grains, and they are localized in the near-surface layer of the film, as shown by the profile of the highlighted surface area (Appendix A). In the spaces between the “technological” strips, the surface is quite smooth. As can be seen from Figure 1a, the change in the height of the surface geometry varies within 2–3 μm.

The surface of the sample heated to 250 °C is strongly textured (Figure 1b and Appendix A) and has a more developed relief than the previous sample. The change in relief height varies in the range of 5–10 μm

The films heated to 300 °C have domain morphology, with elongated elliptical formations tightly adhering to each other on the surface (Figure 1c and Appendix A). The surface of this sample is smoother than the previous one.

During further heating to 350 and 370 °C (Figure 2d,e, Appendix A and Appendix A), the domain morphology is preserved, and the domains decrease in size (contracting effect). At that, the surface geometry irregularities are smoothed out.

Figure 1f,g,h and Appendix A show the results of the AFM study of the hydrolyzed samples. Comparative analysis of AFM images of the surface topography of the PUI polymer film showed that the aggressive media used in the study led to the formation of surface layers of different morphology. Thus, the use of Blend I (Appendix A) in the etching of the film heated to 170 °C resulted in the formation of the least developed relief formed by domains of highly different sizes. At the same time, the domains have clear boundaries, and there are significant gaps between them. When treating the film heated at 300 °C the surface geometry increases (Figure 1g and Appendix A). At the same time, the sample treated in Blend II has a fine domain structure. The sample hydrolyzed in Blend III (Figure 1h and Appendix A) differs from the previous two samples by its largest domain sizes and, by the appearance of the surface profiles, is similar to the sample exposed at 350 °C. Thus, the sample treated with trifluoroacetic acid solution is morphologically more similar to heat-treated samples with a high degree of polycaprolactam block destruction. In general, film hydrolysis preserves the film morphology obtained during annealing but makes it more contrasting due to the destruction of polycaprolactam inclusions.

### 3.2. X-ray Diffraction (XRD) Investigation

Thermolized and hydrolyzed samples were examined by X-ray diffractometry.

The diffractogram from the sample PUI-1-170 °C (PUI, heated at 170 °C) shows one diffuse reflection (amorphous halo) at 2*θ* = 20°, which indicates its amorphous structure. Thermolysis of the copolymer film at 300 °C (curve PUI-1-300 °C in Figure 2, PUI) leads to two additional diffuse reflections at 2*θ* ~ 15° and 27.5°, which form “shoulders” against the main amorphous ring. This indicates that the processes of ordering and formation of crystalline phase nuclei of small size take place in the film. At thermolysis of the sample at 350 °C (curve PUI-1-350 °C in Figure 2, PUI), an increase in reflections intensity at 2*θ* ~ 15° and 27.5° is observed; moreover, the fourth diffuse reflection at 2*θ* ~ 24° and a very weak fifth reflection at 2*θ* ~ 36° appeared in the amorphous halo background which evidences on the perfection of the crystal structure and increases in the crystallites sizes.

Comparison of angular positions of four reflections at 2*θ* ~ 15, 24, 27.5, and 36°, with angular positions of reflections from thermally imidized PMDA-ODA powder (Figure 3, lower curve): 2*θ* ~ 14.9; 22.2, 26.5 (with the shoulder at 27.8) and 36.0°, shows that the formation and growth of aromatic phase crystallites of PMDA-ODA occur during thermolysis at 300–350 °C. No reflections from crystallites of the aliphatic phase of polycaprolactone (Appendix A) were detected on the diffractogram of the PUI-1-350 °C PUI sample (Figure 2). It should be noted that the main very intense and narrow crystal reflections of polycaprolactone at 2*θ* ~ 21.3 and 23.5° (Appendix A) located in the area close to the angle position of the amorphous halo at 2*θ* = 20° were not revealed on the diffractograms of the thermalized samples. 

Hydrolysis of preheated to 170 °C PUI copolymer film in Blend I (curve PUI-1-170 °C-I in Figure 2) does not change its structure. The film remains amorphous because the diffractogram shows only an amorphous halo, as in the sample PUI-1-170 °C.

On the contrary, hydrolysis of PUI copolymer films preliminary thermalized at 300 °C in Blends II and III (curves PUI-1-300 °C-II and UI-1-300 °C-III, respectively) leads to partial structure ordering as diffusion reflections at 2*θ* ~ 15, 24 and 27.5° appear on the diffractograms of hydrolyzed PUI samples, which correspond to the diffraction reflections of the copoly-(urethane-imide) samples heated at 300–350 °C.

Structural studies of copoly(urethane-imides) at different stages of chemical transformations confirm the course of selective destruction of aliphatic (urethane) blocks and their subsequent removal from the near-surface layers of copolymer films during thermal oxidation in the air and hydrolysis in acidic media, which agrees well with the results obtained in our previous work [19]. The experimental data presented in Fig.4 are in accordance with the works [35,36].

### 3.3. Sorption and Filtration Properties of Membranes

The table shows density, degree of swelling, sorption of copoly(urethane-imide) films, as well as their filtration properties. The film thermally treated at 170 °C has the maximum level of swelling in DMF of 38.7% among the investigated samples. This correlates well with our results because, according to TGA and X-ray diffractometry data, the content of polycaprolactam blocks, which are soluble in DMF, is maximal in this sample. Thus, the swelling of the sample is directly related to the content of flexible-chain fragments. Consequently, as the annealing temperature increases, the swelling of the copoly(urethane-imide) films decreases to 20.2% at 300 °C and 5.0 at 350 °C. In this series, the sorption of DMF in the material also decreases from 0.51 g/g at 170 °C to 0.12 g/g at 350 °C. In the case of samples obtained as a result of the combined use of thermal (300 °C) and chemical treatment, both the degree of swelling and sorption of organic solvent are significantly increased (Table 2). This is most noticeable for PUI-1-170 °C-I film heated at 170 °C and treated with Blend I containing 90% of acetic acid. For this sample, the degree of swelling is 33.7%, and sorption is 70%. The rate of swelling of sample PUI-1-170 °C-I is less, and the sorption of solvent is greater than for the sample PUI-1-170 °C heated at 170 °C. This is most likely due to the cleavage of polyester chains and the products of their thermal destruction by acids. In this case, an open porous structure is formed, which absorbs a large amount of solvent. In the case of samples five and six heated at 300 °C, both the rate of swelling and sorption increase. This can be caused by the fact that in the film heated at a high temperature, the polyimide component prevails, which isolates the polycaprolactam blocks from the solvent. The generated pores allow the solvent to penetrate into the sample, which leads to the swelling of polyester chains and increases the sorption of DMF in the material.

It is worth emphasizing that all investigated films are stable in the aprotonic solvent DMF. In this connection, the possibility of filtering through it the solutions of the model substance Remasol Brilliant Blue R was investigated. Films treated at 170–350 °C are practically not permeable by DMF. The exception is the sample PUI-1-170 °C, which shows a very low permeate flux of 0.01 kg/m^2^ h atm and a high retention factor of 95%. This may be due to the fact that the crystalline domains of PI blocks, which are observed on AFM images and recorded by X-ray diffractometry, form a closed porous structure that blocks fluid transfer in the material. In the case of the PUI-1-170 °C sample containing the maximum proportion of caprolactam fragments, the transfer of DMF passes through the swollen polyether phase and not through the pores in the material. This provides a high retention factor but limits the permeate flow.

Films that hydrolyzed in acidic media have completely different transfer properties. As can be seen from Table 2, as a result of the etching of polyether chains, pores of ultrafiltration size 66 μm appear in film four. This increases the permeate flux by more than two orders of magnitude to a value of 2.3 kg/m^2^ h atm. Logically, such a membrane has a low retention factor of 10% for a model pigment with a molecule size of less than 1 μm. The PUI-1-300 °C-II membrane hydrolyzed in Blend II was not permeable in the filtration process. Probably, the etching time was not sufficient to form a thorough porous structure. This correlates well with the sorption data. As can be seen from Table 2, the membrane PUI-1-300 °C-II has the lowest sorption value among all the samples subjected to hydrolysis. Membrane PUI-1-300 °C-III hydrolyzed with trifluoroacetic acid solution has the largest average pore size and permeate flux. At the same time, it practically does not retain the model pigment. At the same time, the lowest swelling in the DMF medium among the hydrolyzed membranes was observed for it. This is associated with greater destruction of polycaprolactone blocks, which is confirmed by X-ray diffractometry data.

## 4. Conclusions

The study corresponds to the rediscovery of poly(4,4′-oxydiphenylene–pyromellitimide) based soluble copoly(urethane-imides) as membrane polymers of a new generation. It is shown that the physical structure of PUI films prepared by the solution method becomes porous after the removal of urethane blocks from the polymer, and the pore size varies depending on the conditions of thermolysis and subsequent hydrolysis of the membrane polymer. It is shown that the surface morphology of the PUI films of the studied samples becomes more prominent as a result of the thermal destruction of polycaprolactam blocks and takes on a domain structure. In general, the hydrolysis of the samples does not significantly change the surface morphology of the films formed during annealing but makes it more contrasting due to the additional destruction of polycaprolactam chains. The film annealed at 170 °C with a low destruction degree of polycaprolactam blocks exhibits the properties of a nanofiltration membrane. It is stable in the aprotic solvent DMF and has a Remasol Brilliant Blue R retention coefficient of 95%. After the hydrolysis of thermally treated films in acidic media, ultrafiltration size 66–82 nm pores appear, which leads to an increase in the permeate flow by more than two orders of magnitude. This circumstance provides opportunities for controlling the membrane polymer structure for further optimization of the performance characteristics of filtration membranes based on it. Thus, we proposed a new preparation method of ultra- and nanofiltration membranes based on poly(4,4′-oxydiphenylene–pyromellitimide) that are stable in aprotic solvents.

As a continuation of the work, it is promising to synthesize PUI with more extended and conformationally hindered polyimide blocks. This will make it possible to form membranes from amorphized polymer with smaller pore size.

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
