# Peer review of "Morphological Analysis of Poly(4,4′-oxydiphenylene-pyromellitimide)-Based Organic Solvent Nanofiltration Membranes Formed by the Solution Method"

_membranes, 2022, doi:10.3390/membranes12121235_

Round 1
Reviewer 1 Report
comment
why the prepared membrane classified as NF membrane?
Author Response
Why the prepared membrane classified as NF membrane?
ANSWER:
The answer to this question follows from the definition of nanofiltration. In this process, the molecular weight of the retained molecules should be 100-1000g/mol.
We studied the process of separating the model substance Remasol Brilliant Blue R dissolved in DMF with a molecular weight of 626 g/mol. The membrane retention coefficient for the model substance was 95%, which reliably corresponds to the parameters of nanofiltration membranes.
Reviewer 2 Report
The authors proposed a new method for fabricating the ultra- and nanofiltration membranes based on poly(4,4'-oxydiphenylene-pyromellitimide) that are stable in aprotic solvents. This study provides the opportunities to control the polymeric structure of membrane for further optimization of the performance characteristics. However, some problems remain in the paper. I recommend that major revision of the manuscript will be required.
(1) In the Abstract, the advantage of the preparation strategy based on poly (4,4'-oxydiphenylene -pyromellitimide) should be clarified, which is important to better understand what the paper described.
(2) There are several formatting errors in the manuscript, including some missing units of quantity and incorrect use of upper and lower marks. It is suggested that the ratio should be replaced by the simplest integer ratio. It is suggested that the table in this paper can be replaced by a three-line table. The arrangement of figures and the mark of figures need to be further improved.
(3) I suggest that the authors should provide the explanation on how the annealed temperature of film was ranged from 170 to 370 °C.
(4) It is suggested to supplement the relevant literatures for further strengthen the analysis of XRD characterization.
(5) I want the author should provide the clear evidence to confirm the formation of selective adsorption on PUI-1-170℃, which is important for preventing fluid transfer in the material.
(6) In addition to AFM and XRD, I suggested that the author should provide some characterizations that can illustrate the main conclusions.
(7) In section “Reference”, unite the forma of journal according to the guidelines. Moreover, there are some grammatical issues that, if corrected, would greatly improve the readability of the paper.
Author Response
Reviewer 2:
- In the Abstract, the advantage of the preparation strategy based on poly (4,4'-oxydiphenylene -pyromellitimide) should be clarified, which is important to better understand what the paper described.
ANSWER:
Thanks for the suggestion. Necessary clarifications are made in the abstract and introduction of the article.
- There are several formatting errors in the manuscript, including some missing units of quantity and incorrect use of upper and lower marks. It is suggested that the ratio should be replaced by the simplest integer ratio. It is suggested that the table in this paper can be replaced by a three-line table. The arrangement of figures and the mark of figures need to be further improved.
- ANSWER:
Formatting errors in the text have been eliminated.
- I suggest that the authors should provide the explanation on how the annealed temperature of film was ranged from 170 to 370 °C.
ANSWER:
In this work, the 170 °C is taken as a temperature that should not be increased to achieve a 100% degree of imidization. We accept that at 170°C, an acceptable degree of imidization of amic acid units (more than 95%) is achieved for us. Thermolysis of urethane links develops above 170°C.
- It is suggested to supplement the relevant literatures for further strengthen the analysis of XRD characterization.
ANSWER:
Two references were added to the list of references 35-36. The text on the X-ray part of the article was corrected.
- I want the author should provide the clear evidence to confirm the formation of selective adsorption on PUI-1-170, which is important for preventing fluid transfer in the material.
ANSWER:
The fact is that the separation in this case is based on a sieve mechanism. Large dye molecules are retained by the membrane, while smaller solvent molecules selectively penetrate through its nanometer-sized pores. Therefore, selective sorption does not affect the separation process. In addition, it should be noted that Poly-(4,4'-oxydiphenylene) pyromellitimide practically does not swell in the separated media, which was one of the main factors for its choice as a membrane material.
- In addition to AFM and XRD, I suggested that the author should provide some characterizations that can illustrate the main conclusions.
ANSWER:
The selective destruction of the synthesized polyurethaneimide was studied in detail in our previous publication using various modern physicochemical methods [Didenko, A.L., Ivanov, A.G.,Smirnova, V.E., Vaganov, G.V., Anokhina, T.S., Borisov, I.L., Volkov, V.V.,Volkov, A.V., Kudryavtsev, V.V. Selective Destruction of Soluble Polyurethaneimide as Novel Approach for Fabrication of Insoluble Polyimide Films. Polymers .2022. 14. 4130. https://doi.org/10.3390/polym14194130].
In this work, the morphology and porous structure of pyrolyzed PUI films were studied not only by AFM and XRD methods, but also by nanofiltration of organic media, as well as by precision capillary flow porosimetry. Therefore, we believe that the main conclusions of the work are experimentally reliably confirmed and substantiated.
- In section “Reference”, unite the forma of journal according to the guidelines. Moreover, there are some grammatical issues that, if corrected, would greatly improve the readability of the paper.
ANSWER:
The list of references has been corrected.
Reviewer 3 Report
The authors examined the selective destruction of co-poly(urethane-imide) films through thermolysis or hydrolysis. Depending on the amount of heat applied, the films exhibited distinct morphologies and can be confirmed byAFM. Upon the treatment, the performance improvement was achieved. Overally, manucript was well written. There are some comments need to be addressed before publication.
1. I need more explanation about “technological surface geometry” at 230th row.
2. Why is the surface of the heated film at 250 °C rougher than the surface of the heated film at 170 °C? Clear explanation is necessary.
3. A high-resolution XRD pattern is required. (Currently, the difference is not clear.)
4. Upon heating the polymer membrane, the polymer chain could be affected. An IR or Raman result may be required.
Author Response
- I need more explanation about “technological surface geometry” at 230th row.
ANSWER:
We propose in the proofreading text to replace the term "technological surface geometry" with the term "technological relief".
- Why is the surface of the heated film at 250 °C rougher than the surface of the heated film at 170 °C? Clear explanation is necessary.
ANSWER:
In the temperature range of 170-250 ° C, several processes occur, accompanied by the release of volatile products, namely, the completion of cyclization of residual amic acid units in the polymer, thermal oxidative decomposition of urethane units, removal of residues of catalytic systems. As a result, there is a local swelling of the near-surface layers of the polymer and the formation of pores and cavities on the surface of the films, which leads to an increase in the values of the surface roughness parameters.
- A high-resolution XRD pattern is required. (Currently, the difference is not clear.) ANSWER:
The authors agree that the resolution of the diffractograms in Figure 3 is rather low. This may be due to the thickness of the polymer film, which is less than 100 µm. However, it was important for us to study by X-ray exactly those samples that were then tested in nanofiltration. Despite the fact that the diffraction patterns cannot be characterized quantitatively, the appearance of reflections at 15° and 27.5° during heat treatment, corresponding to the crystalline phase of polyimide, can be reliably recorded. It is also reliably shown that with an increase in the film treatment temperature from 170 to 350°C, the intensity of these reflections increases.
- Upon heating the polymer membrane, the polymer chain could be affected. An IR or Raman result may be required.
ANSWER:
We have inserted IR spectra and their explanation into the supplementary.
Round 2
Reviewer 2 Report
The author has provided good answers to all the questions, and the manuscript has been revised according to the comments. Therefore, it can be accepted for publication in Membranes.